# Nanostructured Polyelectrolyte Complexes Based on Water-Soluble Thiacalix[4]Arene and Pillar[5]Arene: Self-Assembly in Micelleplexes and Polyplexes at Packaging DNA

**DOI:** 10.3390/nano10040777

**Published:** 2020-04-17

**Authors:** Luidmila S. Yakimova, Aigul R. Nugmanova, Olga A. Mostovaya, Alena A. Vavilova, Dmitriy N. Shurpik, Timur A. Mukhametzyanov, Ivan I. Stoikov

**Affiliations:** A.M. Butlerov Chemical Institute, Kazan Federal University, Kremlevskaya Street, 18, 420008 Kazan, Russia; aygul9pul9@mail.ru (A.R.N.); olga.mostovaya@mail.ru (O.A.M.); anelia_86@mail.ru (A.A.V.); DNShurpik@mail.ru (D.N.S.); timmie.m@gmail.com (T.A.M.)

**Keywords:** nanomaterials, thiacalix[4]arene, pillar[5]arene, interpolyelectrolyte, co-assembly, selective recognition, DNA packing

## Abstract

Controlling the self-assembly of polyfunctional compounds in interpolyelectrolyte aggregates is an extremely challenging task. The use of macrocyclic compounds offers new opportunities in design of a new generation of mixed nanoparticles. This approach allows creating aggregates with multivalent molecular recognition, improved binding efficiency and selectivity. In this paper, we reported a straightforward approach to the synthesis of interpolyelectrolytes by co-assembling of the thiacalix[4]arene with four negatively charged functional groups on the one side of macrocycle, and pillar[5]arene with 10 ammonium groups located on both sides. Nanostructured polyelectrolyte complexes show effective packaging of high-molecular DNA from calf thymus. The interaction of co-interpolyelectrolytes with the DNA is completely different from the interaction of the pillar[5]arene with the DNA. Two different complexes with DNA, i.e., micelleplex- and polyplex-type, were formed. The DNA in both cases preserved its secondary structure in native B form without distorting helicity. The presented approach provides important advantage for the design of effective biomolecular gene delivery systems.

## 1. Introduction

Innovative technologies inspired by self-assembly offer promises in the creation of the nanoscale architectures for constructing intelligent materials, molecular machines, reading devices, electromechanical keys, and sensors [1,2,3]. Various self-assembly processes lead to spontaneous organization of small molecules into large well-defined stable supramolecular aggregates, which are useful in numerous applications [4]. Association of oppositely charged macromolecules (polyelectrolytes) leads to the formation of interpolyelectrolyte complexes (IPEC) [5,6]. These materials are applied in encapsulation [7,8] and therapeutics delivery [9,10]. Despite the fact that the complexation between linear polyelectrolytes has been carefully studied [11,12,13], utilizing the multifunctional macrocycles with charged fragments opens new possibilities for constructing self-assembled complexes. Using preorganized macrocycles allows creating clearly defined hierarchical structures (spheres or cylinders, fractal type, etc.) unlike linear macromolecules, which as a rule of thumb form spherical particles [11,12,13]. Additional control of the structure and stability of such complexes can be achieved by varying the structure and composition of a macrocycle. Therefore, utilizing preorganized macrocycles is an effective strategy in managing the structure and properties of IPEC.

For the synthesis of co-interpolyelectrolyte, we have chosen thiacalix[4]arene and pillar[5]arene. Both macrocycles belong to the class of cyclophanes. Meanwhile, pillar[5]arenes are 1,4-disubstituted paracyclophanes while thiacalix[4]arenes are 1,3-disubstituted metacyclophanes. Different linkage of aromatic rings by bridges (methylene or sulfide groups) provides different shapes of macrocycles, i.e., tube in the case of pillar[5]arene and cone in the case of thiacalix[4]arene. Besides, these macrocycles differ in the number and spatial arrangement of the functional groups. Thus, thiacalix[4]arenes contain four substituents at the lower rim while pillar[5]arenes contain five functional groups on each side of the macrocycle. Thiacalix[4]arene and pillar[5]arene platforms are used to create self-organized molecular monolayers [14,15,16,17], nanotubes [18], artificial ion channels [19], molecular devices [20], and nanomaterials [21]. The formation of host-guest complexes is typical for macrocyclic structure of calixarenes [22,23,24,25,26]. Meanwhile, pillar[5]arenes as paracyclophanes can include hydrocarbon fragments in their cavity. This leads to self-assembly with formation of mechanically interlocked molecules and nanomachines: rotaxanes, polyrotaxanes, pseudorotaxanes, and catenanes [27,28,29,30]. In this regard, the simultaneous use of two such completely different macrocycles may lead to the creation of principally new supramolecular structures with unique properties different from those of individual species. Thus, molecular design of interpolyelectrolyte aggregates via combination of a number of oppositely charged polyionic macrocyclic platforms can be considered as a new direction in supramolecular chemistry and material science [31,32,33]. For the first time, the use of both pillar[5]arene and thiacalix[4]arene representatives as components of self-assembled systems has been proposed. This approach allows obtaining mixed self-assembled structures both in solution and in solid phase. Macrocycle-based interpolyelectrolyte complexes have several important features: increased selectivity of the material obtained toward small analytes, stabilization of the adsorbed enzyme and DNA biomolecules providing their comfortable microenvironment, and decrease in nonspecific sorption of proteins by varying morphology of nanoparticles. To date, no such homogeneous and heterogeneous mixed self-assembled structures had been previously developed for recognition and separation of the biopolymers. 

Here, we propose a new supramolecular approach to the synthesis of co-interpolyelectrolytes based on water-soluble sulfonated thiacalix[4]arene (STC[4]A) and ammonium derived pillar[5]arene (AP[5]A) combined into one ensemble (Figure 1). Mixed oppositely charged macrocycles can spontaneously assemble into supramolecular aggregates, e.g., interpolyelectrolyte nanoparticles, to provide multiple binding sites on their surface. Such self-assembled interpolyelectrolyte nanoparticles were obtained for the first time. Recognition and complexation of polycation AP[5]A and co-interpolyelectrolyte aggregate AP[5]A/STC[4]A with high-molecular DNA from calf thymus were studied.

## 2. Materials and Methods 

### 2.1. Sample Preparation

Interpolyelectrolyte nanoparticles were prepared by mixing two oppositely charged compounds in solution (AP[5]A + STC[4]A) in different molar ratios (1:1, 1:2, 1:2.5, 1:3, and 1:9). Initial concentration was equal to 3 × 10^−4^ M, final volume of the mixture was 1 mL. For experiments with dilution, each subsequent concentration was prepared by 10-fold dilution of a more concentrated previous solution of co-interpolyelectrolyte aggregates.

### 2.2. Scanning Electron Microscopy (SEM)

SEM analysis was carried out using the Carl Zeiss Auriga Cross Beam scanning electron microscope (Oberkochen, Germany). For the sample preparation, 10 μL of the solution were placed on the silica substrate, which was then dried at room temperature during 1 h.

### 2.3. Dynamic Light Scattering (DLS)

#### 2.3.1. Particles’ Size 

The particles size was determined by the Zetasizer Nano ZS instrument (Worcestershire, UK) at 20 °C. The instrument was equipped with the 4 mW He-Ne laser operating at a wavelength of 633 nm and incorporated noninvasive backscatter optics (NIBS). Measurements were performed at the detection angle of 173° with the software automatically determining the measurement position within the quartz cuvette. Each experiment was repeated three times. Macrocycles (STC[4]A and AP[5]A) dissolved completely in water at the concentrations used in the work (from 3 × 10^−6^ M to 3 × 10^−4^ M). 

#### 2.3.2. Zeta Potentials 

Zeta (ζ) potentials were measured on a Zetasizer Nano ZS from Malvern Instruments. Samples were prepared as for the DLS measurements and were transferred with the syringe to the disposable folded capillary cell for measurement. The zeta potentials were measured using the Malvern M3-PALS method and averaged from three measurements.

### 2.4. UV-Visible Spectroscopy

UV-visible spectra were recorded on the Shimadzu UV-3600 spectrophotometer (Kyoto, Japan) using 1 cm quartz cuvette at 25 °C. Calf thymus DNA sodium salt (CT-DNA) (Sigma; sodium content 6%) was used as received. Concentration of the CT-DNA solution was determined using ε_260_ = 6600 mol^−1^cm^−1^ expressed in the base pair equivalents per liter. Purity of the DNA was checked by the ratio of the absorbance A_260_/A_280_ > 1.8, indicating the DNA was sufficiently free from protein. Two mL of 2.4 × 10^−4^ M CT-DNA solution were added to 0.1 mL of the host solution (interpolyelectrolyte aggregates or the macrocycle AP[5]A, 3.4 × 10^−3^ M) in 10 mM Tris-HCl containing, 10 mM NaCl, pH = 7.4, and diluted to final volume of 3 mL. Then UV spectra of the solution were recorded. 

### 2.5. Fluorescence Spectroscopy

The 4,6-diamidino-2-phenylindole (DAPI) fluorescence spectra were recorded on the Fluorolog 3 luminescent spectrometer (Horiba Jobin Yvon, Kyoto, Japan) at an excitation wavelength of 347 nm and emission scan range of 400‒530 nm. Excitation and emission slits were equal to 2 nm for DAPI in the presence of AP[5]A and to 1 nm in the presence of AP[5]A/STC[4]A. Quartz cuvettes with an optical path length of 10 mm were used. The cuvette was located in the front face position. Emission spectra were automatically corrected using the Fluorescence program. The fluorescence spectra were recorded with the 6 µM concentration for DAPI and CT-DNA and 6 µM—for AP[5]A and AP[5]A/STC[4]A. The experiments were carried out at 20 °C. Solutions of the investigated systems were measured after incubating for an hour at room temperature.

### 2.6. Circular Dichroism (CD) Studies

The changes in the intensity of the CD signal of CT-DNA alone and in the presence of interpolyelectrolyte aggregates AP[5]A/STC[4]A or macrocycle AP[5]A were recorded from 245 nm to 310 nm at 25 °C using a Jasco J-1500 spectropolarimeter (Tokyo, Japan) in a quartz cuvette with a 10 mm optical path length. The 2.4 × 10^−4^ M solution of the CT-DNA 2 mL was added to 0.1 mL of the solution of host (interpolyelectrolyte aggregates or macrocycle AP[5]A) (3.4 × 10^−3^ M) in aqueous buffer (10 mM Tris-HCl, 1 mM NaCl, pH = 7.4) and diluted to final volume of 3 mL. 

## 3. Results

### 3.1. Construction of the AP[5]A/STC[4]A Co-Assemblies 

To create mixed systems based on the self-assembly of water-soluble charged macrocycles, cationic pillar[5]arenes and anionic thiacalix[4]arenes were chosen. The AP[5]A, positively charged macrocycle, contained 10 quaternary ammonium fragments, and an oppositely charged macrocycle (thiacalix[4]arene STC[4]A) in cone conformation was functionalized at the lower rim with four alkyl sulfonate fragments (Figure 1). 

Mixed aggregates were self-assembled by mixing aqueous polycation and polyanion solutions. The total concentration of the macrocycles in the solution was equal to 3 × 10^−4^ M while the ratio of the components mixed was varied. The thiacalix[4]arene contained four negatively charged functional groups at the one side of the cavity whereas pillar[5]arene contained 10 ammonium groups located at both sides. Therefore, it was necessary to find the ratio of the macrocycles leading to the most stable monodisperse system of co-interpolyelectrolyte nanoparticles. For this goal, the series of experiments with different molar ratio of AP[5]A:STC[4]A was performed, where the macrocycle STC[4]A was taken in an excess. The 1:1, 1:2, 1:2.5, 1:3, and 1:9 molar ratios were chosen (Appendix A). To control the formation of mixed nanoparticles, self-assembly of each component (AP[5]A and STC[4]A) was investigated in water by DLS, a method providing hydrodynamic diameter of the nanoparticles and self-associates as well as the stability and charge of the particles formed according to their ζ-potential. Only two systems (1:2 and 1:2.5) from those studied showed formation of monodisperse system (Appendix A). This indicated the stoichiometry of the co-interpolyelectrolyte nanoparticles was close to 1:1 (monomolar ratio of charged groups). AP[5]A/STC[4]A co-assemblies in 1:2 ratio had hydrodynamic diameter of 129 nm, PDI = 0.18 and ζ-potential of + 37 mV. The AP[5]A/STC[4]A co-assembled in 1:2.5 ratio had hydrodynamic diameter of 158 nm, low PDI, and ζ-potential of +35 mV. All these features, including rather narrow size distribution and high ζ-potential, clearly demonstrated the formation of stable colloidal system. Deficiency and excess of STC[4]A against the ratio specified above resulted in a worse charge compensation and caused an increase in polydispersity of the solution and hydrodynamic diameter of the aggregates (Appendix A). 

### 3.2. Co-Assembly with DNA 

DNA from calf thymus (CT-DNA) was chosen as model substrate. We set out to explore the interaction of the synthesized AP[5]A/STC[4]A co-assembly with positively charged surface, and the polyanionic DNA by UV, CD, and fluorescence spectroscopy. Pillar[5]arene AP[5]A as a component of co-interpolyelectrolyte associate AP[5]A/STC[4]A contained 10 ammonium fragments, which could also interact with the CT-DNA. Based on the results on stability and monodispersity of the co-interpolyelectrolyte aggregates AP[5]A/STC[4]A given above, the colloid solutions with ratio of AP[5]A/STC[4]A 1:2 were taken. The same experiment was performed between AP[5]A and CT-DNA.

#### 3.2.1. UV Spectroscopy

The ability to interact with DNA was determined by increasing or decreasing the maximum absorption and band shift. The interaction of the AP[5]A/STC[4]A ligand with the CT-DNA did not lead to any changes in the UV spectrum, which indicated absence of the conformational changes in the DNA structure (Appendix A). Only noncovalent interaction due to electrostatic binding to negatively charged ribose phosphate residues of the DNA backbone was observed. Similar results were obtained in the case of the interaction of the macrocycle AP[5]A with CT-DNA (Appendix A).

#### 3.2.2. Fluorescence Spectroscopy 

In order to study the interaction of AP[5]A and AP[5]A/STC[4]A with the CT-DNA, fluorescence spectroscopy was used (Figure 2). We chose DAPI as a common fluorescent marker of DNA interactions that was also used as antiparasitic, antibiotic, antiviral, and anticancer drug.

At low DAPI/DNA ratio, DAPI interacted with DNA preferably due to binding to A–T pairs [35,36]. An increase in the dye concentration led to the additional possibility of binding in minor grooves due to phosphate groups of DNA. The character of the binding could be easily controlled by adjusting the ratio of the dye and biopolymer [37]. Thus, we chose high concentration of the dye (1 dye molecule per DNA base pair), resulting in binding both in the minor groove and intercalation [38]. The interaction of the DAPI/CT-DNA system with the AP[5]A was first studied. In the presence of the macrocycle, the dye fluorescence intensity increased against blank experiment. This is not typical for such systems [39,40]. A slight shift of the emission maximum to the red region of the spectrum was observed (from 450 to 456 nm) (Figure 2A). 

Such a shift in the emission maximum resulted from the preferential binding of both the dye and paracyclophane on the phosphate groups of DNA. Polarity of the medium surrounding the dye was increased due to the charge of the macrocycle [41]. The DNA molecule had an extremely high negative charge density, which could not be shielded by buffer components [42]. As a result, electrostatic binding of positively charged AP[5]A with the phosphate groups of the DNA was observed. Thus, the increase in the emission intensity can be associated with an additional shielding of the dye against the buffer by the micelles. In addition, no change in the emission spectrum of free DAPI in the presence of AP[5]A was observed (Figure 2A). Most likely, positively charged dye could not be incorporated into the micelles formed by similarly charged cyclophane. All of the data confirmed that the AP[5]A bound to the DNA. 

Interestingly, the DAPI fluorescence intensity significantly increased in the presence of AP[5]A/STC[4]A interpolyelectrolyte particles carrying positive charge (Figure 2B). Such an effect can result from the formation of a highly unstable system [43] at low particle concentrations. In this case, the dye molecules had chance to incorporate into the micelles. Actually, DLS data showed the formation of an unstable system AP[5]A/STC[4]A at 3 × 10^−6^ M (ζ-potential is +10 mV) (Appendix A). In the DNA presence, increase in the fluorescence of the DAPI + CT-DNA + [AP[5]A/STC[4]A] system was more distinct. The character of the changes was completely analogous to that of the DAPI + CT-DNA + AP[5]A system. This suggested similar interaction model, i.e., binding of the interpolyelectrolyte particles to phosphate groups of the biopolymer.

#### 3.2.3. Circular Dichroism Spectroscopy 

The conformational changes in CT-DNA structure due to ligand (AP[5]A/STC[4]A and AP[5]A) binding can be also accessed using CD spectroscopy (Figure 3).

CD spectrum of CT-DNA had a positive signal at 275 nm, which corresponded to base stacking and a negative signal at 245 nm attributed to right-handed helicity, which indicated a mixture of A and B forms of DNA [44,45,46,47]. Canonical A and B forms of DNA were simultaneously present in aqueous solutions. Conversion of A-DNA into B-DNA in aqueous solution occurs in mild conditions depending on the environment [48,49]. In the presence of AP[5]A/STC[4]A and AP[5]A, a 13-nm red shift of the band at 275 nm was observed. These spectral changes confirmed the preservation of the canonical form of DNA. However, a slight change in the ratio of A and B forms was observed, with an increase of the A form in the content [50]. These changes indicated that CT-DNA retained a B form upon binding, but at the same time underwent some conformational changes. Therefore, CT-DNA was bound by ligand (AP[5]A/STC[4]A and AP[5]A) due to electrostatic interactions with phosphate groups. The impossibility of intercalating the studied compounds in DNA confirmed the preservation of the shape of the CD curve: The absence of additional bands in the spectrum indicated the noncleavage of the biopolymer molecule [50], which was in good agreement with the data obtained by fluorescence spectroscopy (Figure 2). 

#### 3.2.4. Dynamic Light Scattering

The baseline increase in the UV spectra was observed during the interaction of the AP[5]A/STC[4]A with the CT-DNA (Appendix A) due to the association of the DNA with the AP[5]A/STC[4]A [51]. Furthermore, the solutions of the biomacromolecules were not able to scatter light (Rayleigh scattering). Thus, they could not exert artificially increased absorption. The data obtained demonstrated the association of the synthetic macrocycle AP[5]A and co-interpolyelectrolyte aggregates AP[5]A/STC[4]A with the model DNA. According to the results, we would like to confirm an interaction of a biopolymer with receptors that led to the formation of aggregates and to estimate their size by DLS. To confirm the association of the macrocycles with DNA, the size of the particles based on the AP[5]A/STC[4]A+CT-DNA was determined, and self-association of CT-DNA and co-interpolyelectrolyte aggregates studied. Varying the ratio of the aggregates and the DNA molecules, the best DNA packaging was observed at the component ratio of 1:3 (Table 1, Appendix A). Note that the CT-DNA in Tris-HCl had the hydrodynamic diameter of 2612 nm [52]. 

The unimodal distributions were observed for these systems (Table 1, Appendix A). The ζ-potentials of these aggregates had negative value, which suggests that the surface of the positively charged particles was covered with polyanionic DNA molecules (Appendix A). The AP[5]A/STC[4]A formed a stable complex with the CT-DNA according to their ζ-potentials. The slight decrease in the ζ-potential was observed for the aggregates [AP[5]A/STC[4]A]+CT-DNA with decreasing of molar ratio. In this case, the associate was obviously compacted. The DLS showed an efficient packing of high-molecular CT-DNA molecules by the AP[5]A to the size of 109 nm (Figure 4). The charge of the aggregates formed in the process of the CT-DNA packaging was positive, +24 mV (Figure 4).

Based on the correlation between magnitudes and signs of the ζ-potential for the aggregates formed by co-interpolyelectrolyte AP[5]A/STC[4]A and self-associates of the macrocycle AP[5]A, we proposed their different structures (Figure 5). 

When the co-interpolyelectrolyte associate [AP[5]A/STC[4]A] and DNA were mixed, the surface charge of the formed aggregates changed its sign from positive to negative. This indicated that the surface of the aggregates was covered with the DNA molecules. When the associate was formed by the DNA addition to the macrocycle AP[5]A, the surface charge was positive. This is possible when the AP[5]A electrostatically interacts with the phosphate residues of the DNA molecule which wraps them as necklace. Both cases resulted in significant packaging of DNA.

In the case of the AP[5]A/DNA aggregates, the DNA chain was protected from contact with the external medium. In particular, this protected DNA from digestion by nucleases [53]. Meanwhile, these complexes can undergo drastic rearrangements and release DNA during their interaction with those components of the cell membrane that are capable to compete for cooperative binding with polycations [54].

#### 3.2.5. Scanning Electron Microscopy

To confirm proposed structures of the CT-DNA aggregates based on the co-interpolyelectrolyte aggregate AP[5]A/STC[4]A and the polycation based on the macrocycle AP[5]A, scanning electron microscopy was used (Figure 6, Appendix A).

As revealed by SEM images shown in Figure 6, two types of the nanostructured aggregates formed by the AP[5]A and the AP[5]A/STC[4]A were considerably different in their packaging though they involved similar cationic blocks AP[5]A (Figure 6A–D). While AP[5]A formed cube-like aggregates with the edge length of about 200 nm (Figure 6A), co-interpolyelectrolyte aggregate AP[5]A/STC[4]A (Figure 6C) had nearly globular shape (sphere and oval) of a larger size. The addition of the CT-DNA to the polycation AP[5]A led to DNA compaction in the cube structure (Figure 6B). In contrast, AP[5]A/STC[4]A aggregates gave beads-on-a-string structures [55,56,57], where the CT-DNA chains wrapped around and bridged these particles (Figure 6D). In case of linear polycations and IPEC, these types of aggregates with the DNA are called polyplexes and micelleplexes [53]. Micelleplexes formed by co-interpolyelectrolyte aggregates AP[5]A/STC[4]A and DNA molecules due to surface covering by DNA can be used as matrices for assembling molecular nanostructures, e.g., proteins and peptides encoding different information (molecular lithography). Besides, DNA wrapped around the micelle corona may be more accessible to transcription of enzymes and unpackaging/accessibility could be an additional reason for higher transgene expression [55,56,57].

### 3.3. Determination of the Proposed Structure of Co-Interpolyelectrolyte Aggregates AP[5]A/STC[4]A 

In order to determine the structure of the interpolyelectrolyte complexes (Figure 5), in particular, possible mechanism for packing the macrocycles AP[5]A and STC[4]A in the structure of the complex, DLS experiments were performed with the complex dilutions. The determined hydrodynamic diameters of the particles are presented in Table 2. Previously, we also studied the self-assembly of initial macrocycles (Table 2). STC[4]A with negatively charged sulfonate groups gave the system with rather low polydispersity indexes within the concentration range studied. Submicron-sized aggregates (d = 168 nm and PDI = 0.32) were formed at 3 × 10^−5^ M [34]. According to the polydispersity index, these distributions were polymodal in a wide range of concentrations of the macrocycle AP[5]A. AP[5]A formed unstable polydisperse systems within several size ranges. The main part of AP[5]A assemblies had a hydrodynamic diameter from 167 up 200 nm, high polydispersity index (PDI), and a positive surface potential of +14 mV (Appendix A). AP[5]A/STC[4]A co-assemblies in 1:2 ratio at 3 × 10^−4^ M had a hydrated diameter of 129 nm, PDI = 0.18, and ζ-potential of +37 mV. All these features, including a rather narrow size distribution and a well-defined ζ-potential, clearly demonstrated the formation of a uniform and homogeneous AP[5]A/STC[4]A co-interpolyelectrolyte nanoparticles. 

When colloidal system AP[5]A/STC[4]A was diluted from 3 × 10^−4^ to 3 × 10^−6^ M, the ζ-potential decreased (Table 2, Appendix A). Such a decrease of surface charge clearly indicated the desorption of the potential-determining ions from the surface of the dispersed phase [58]. Based on the correlation between the structure of the pillar[5]arene and early studied thiacalix[4]arene in *1,3-alternate* conformation [33], such an effect is explained by packing macrocycles in the structure of co-interpolyelectrolyte nanoparticles. Pillar[5]arene, as well as thiacalix[4]arene in 1,3-alternate conformation, had functional groups on the both rims of the macrocycle. Schematic representation of the supposed packing of the macrocycles in various conformations in the composition of polyelectrolyte aggregates AP[5]A/STC[4]A is shown in Figure 7. 

Macrocycles AP[5]A and STC[4]A in the polyelectrolyte aggregates AP[5]A/STC[4]A were interconnected via electrostatic and hydrophobic interactions. On the base of our work, STC[4]A can form micelles at the critical micelle concentration of 3 × 10^−5^ M [34]. Macrocycle STC[4]A formed the core of the spherical co-interpolyelectrolyte nanoparticles. Oppositely charged AP[5]A was assembled on micelle surface due to electrostatic and hydrophobic interactions. Bilayer structures were formed. With diluting of the solution of co-interpolyelectrolyte aggregates AP[5]A/STC[4]A, some layers (including bilayers) could be removed (Figure 7). The number of leached layers was a multiple of three. For this reason, the surface was always positively charged. The dilution properties of co-interpolyelectrolyte aggregates AP[5]A/STC[4]A allowed concluding that such layer-by-layer disassembly led to formation of aggregates with different size and of the surface (Table 2). When system AP[5]A/STC[4]A was diluted from 3 × 10^−4^ to 3 × 10^−5^ M, the hydrodynamic diameter slightly decreased from 129 to 124 nm. This might have been due to removal of several molecular layers. With further dilution up to 3 × 10^−6^ M, the associate size increased. Probably, when the solution was diluted, the mobility of the polyelectrolyte molecules of the outer layer increased and colloid aggregation took place (Figure 7).

## 4. Conclusions

In conclusion, we demonstrated a straightforward method for the synthesis of nanostructured co-interpolyelectrolyte aggregates formed by two macrocyclic species differing in the number, type, and spatial location of functional groups, i.e., thiacalix[4]arene and pillar[5]arene. The capacity of these co-interpolyelectrolyte aggregates to form stable nanoparticles with the CT-DNA was confirmed by the UV-VIS, CD and fluorescence spectroscopy, DLS, and SEM. Varying the aggregates/DNA ratio, the best DNA packaging was achieved at a component ratio of 1:3. The SEM experiment showed two types of aggregates (polyplex and micelleplex) formed by the AP[5]A and the AP[5]A/STC[4]A even though they included similar cationic blocks of AP[5]A. The DLS method showed an efficient packing of the high-molecular CT-DNA to the nanosized aggregates of 109 and 186 nm using the AP[5]A and the AP[5]A/STC[4]A, correspondingly. The CT-DNA packaging by polycationic AP[5]A and co-interpolyelectrolyte aggregates AP[5]A/STC[4]A preserved the DNA secondary structure in its native B form without distorting its helicity. This is an important advantage for the design of effective biomolecular delivery systems. On the basis of the results obtained, we proposed that the AP[5]A and STC[4]A macrocycles can be considered as useful tools for construction of the ensembles with the DNA molecules either entrapped in the internal space of micelleplex (in case of co-interpolyelectrolyte aggregates AP[5]A/STC[4]A) or bound on their surface (in case of AP[5]A). DNA was here used not only as a component of nanostructured material, but also as a matrix for assembling nanostructures. Therefore, micelleplexes can be used as matrices for assembling molecular nanostructures (molecular lithography) and for biomedical applications. The main application of polyplexes is DNA banking, i.e., secure, long-term storage of human genetic material.

## Figures and Tables

**Figure 1 nanomaterials-10-00777-f001:**
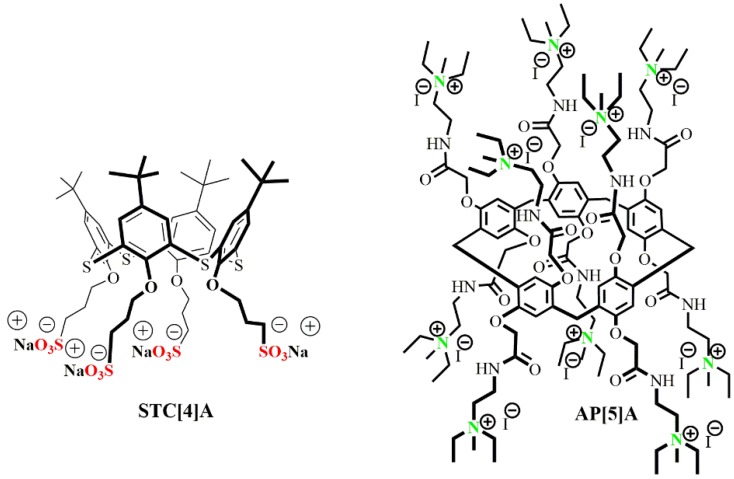
Structures of the thiacalix[4]arene STC[4]A and pillar[5]arene AP[5]A derivatives applied as components of co-interpolyelectrolyte nanoparticles. These compounds were synthesized in a previous study [30,34]. Adapted from [34], with permission from publisher Springer Nature, 2020.

**Figure 2 nanomaterials-10-00777-f002:**
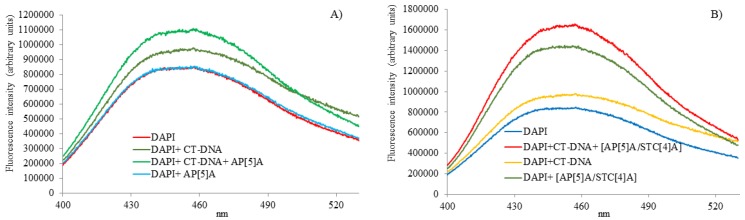
(**A**) The effect of the AP[5]A (6 μM) and (**B**) of the AP[5]A/STC[4]A (6 μM) on DAPI fluorescence spectra (6 μM) in the absence and presence of the CT-DNA (6 μM) in 10 mM Tris-HCl buffer (pH 7.4).

**Figure 3 nanomaterials-10-00777-f003:**
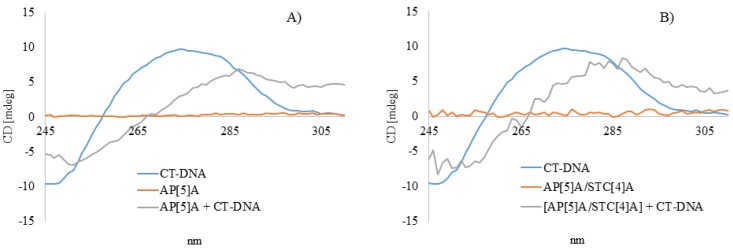
(**A**) CD spectra for AP[5]A, CT-DNA, and AP[5]A+CT-DNA system in 1:3 molar ratio and (**B**) CD spectra for AP[5]A/STC[4]A, CT-DNA, and AP[5]A/STC[4]A+CT-DNA system in 1:3 molar ratio. The concentration of CT-DNA is 1.6 × 10^−4^ M.

**Figure 4 nanomaterials-10-00777-f004:**
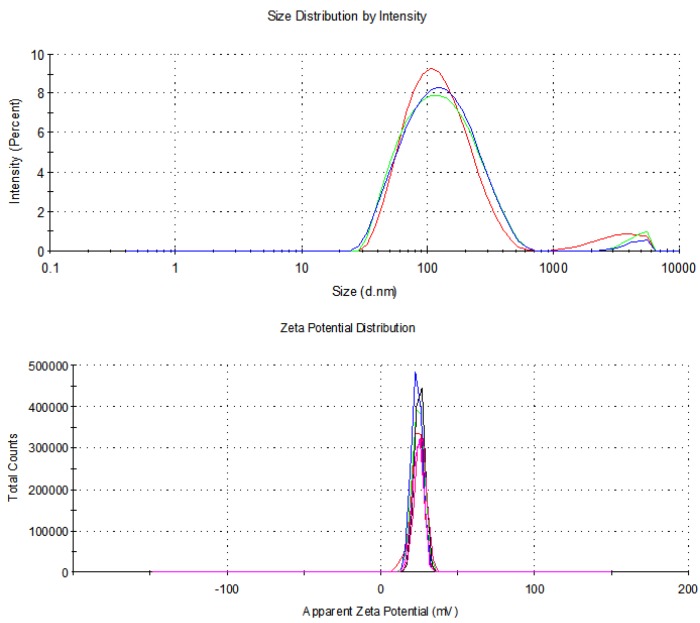
Size and zeta-potential distributions of the AP[5]A + CT-DNA aggregates taken in the 1:3 molar ratio. CT-DNA concentration was 0.9 × 10^−4^ M.

**Figure 5 nanomaterials-10-00777-f005:**
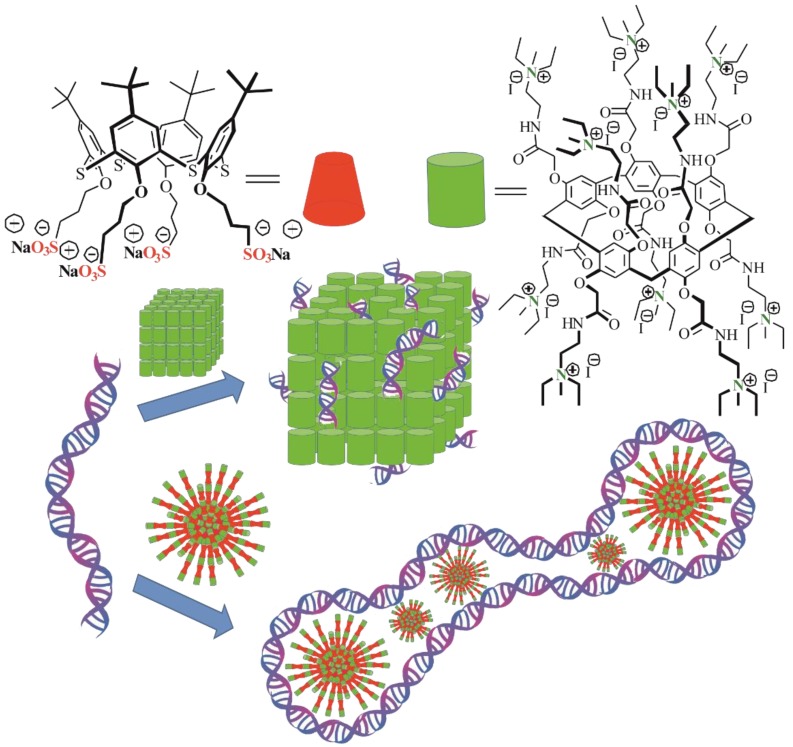
Proposed structures of the CT-DNA aggregates based on the co-interpolyelectrolyte associate AP[5]A/STC[4]A and the polycation based on macrocycle AP[5]A.

**Figure 6 nanomaterials-10-00777-f006:**
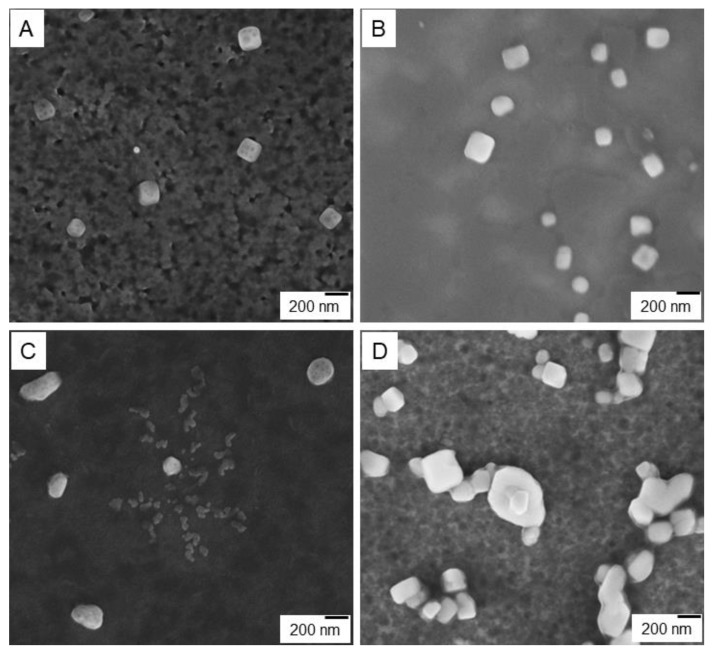
SEM images: (**A**) AP[5]A aggregate, (**B**) AP[5]A/CT-DNA polyplex, (**C**) co-interpolyelectrolyte aggregates AP[5]A/STC[4]A, and (**D**) [AP[5]A/STC[4]A]+CT-DNA micelleplex.

**Figure 7 nanomaterials-10-00777-f007:**
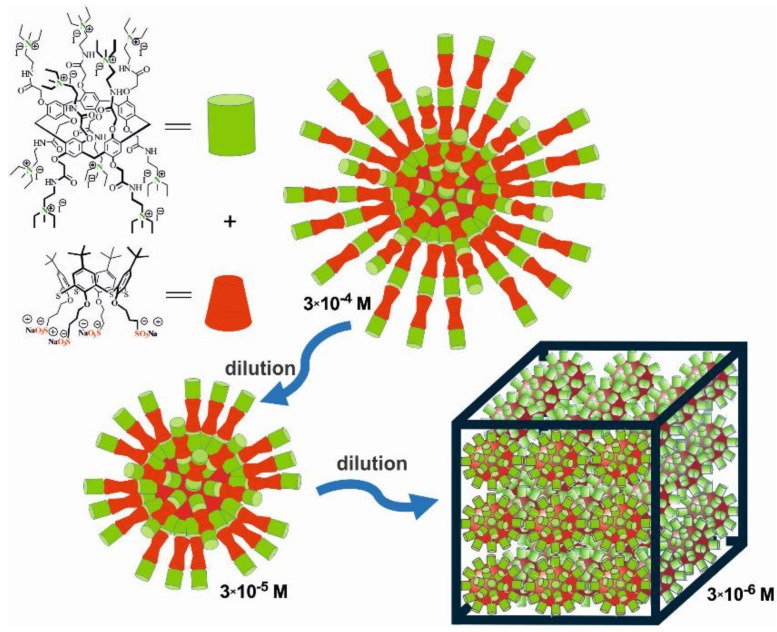
Schematic representation of the structure of the interpolyelectrolyte aggregates at 1:2 molar ratio between anionic STC[4]A and cationic AP[5]A macrocycles during layer-by-layer disassembly upon dilution.

**Table 1 nanomaterials-10-00777-t001:** The size of aggregates (hydrodynamic diameters of particles d, nm) formed by AP[5]A/STC[4]A with CT-DNA in Tris buffer, polydispersity index (PDI), ζ-potential.

Molar Ratio	[AP[5]A/STC[4]A]+CT-DNA		
	PDI	d, nm	ζ, mV
1:10	0.32 ± 0.04	199 ± 2	−44 ± 3
1:7	0.30 ± 0.01	192 ± 2	−41 ± 2
1:3	0.23 ± 0.01	186 ± 1	−36 ± 1

**Table 2 nanomaterials-10-00777-t002:** The size of aggregates (hydrodynamic diameters of particles d, nm) formed by STC[4]A, AP[5]A and AP[5]A/STC[4]A co-interpolyelectrolyte (1:2 molar ratio) in water, polydispersity index (PDI), ζ-potential.

c,* mol/L	STC[4]A [34]		AP[5]A		AP[5]A/STC[4]A		
	PDI	d, nm	PDI	d, nm	PDI	d, nm	ζ, mV
3 × 10^−4^	0.41 ± 0.07	480 ± 73	0.40 ± 0.22	183 ± 12	0.18 ± 0.01	129 ± 5	+37 ± 1
3 × 10^−5^	0.32 ± 0.02	168 ± 8	0.45 ± 0.19	167 ± 54	0.27 ± 0.02	124 ± 4	+16 ± 1
3 × 10^−6^	0.38 ± 0.04	262 ± 25	0.35 ± 0.07	200 ± 38	0.40 ± 0.06	154 ± 19	+10 ± 1

* In case of co-interpolyelectrolyte nanoparticles AP[5]A/STC[4]A, this concentration corresponds to concentration of AP[5]A in the associate AP[5]A/STC[4]A.

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
