# Peer review of "Nanostructured Polyelectrolyte Complexes Based on Water-Soluble Thiacalix[4]Arene and Pillar[5]Arene: Self-Assembly in Micelleplexes and Polyplexes at Packaging DNA"

_nanomaterials, 2020, doi:10.3390/nano10040777_

Round 1

Reviewer 1 Report

This manuscript by Yakimova and coworkers reports the synthesis of polyelectrolytes by self-assembly of a positively charged pillar[5]arene and a negatively charged calix[4]arene. The interaction of the systems obtained and DNA is studied using absorption and emission spectroscopy. The authors find that the behaviour of the polyelectrolytes with DNA is different than the behaviour of the pillar[5]arene with DNA.

This paper can be potentially of interest to supramolecular chemists and researchers interested in biopolymer materials. I say potentially because in my opinion the structure and the language used in the manuscript make it difficult to follow by non-specialist readers. It is a shame, since if the manuscript was written in a more friendly manner it will be more accessible to these audiences. I have expertise in many of the techniques (absorption, emission, circular dichroism) that the authors use and in the study of self-assembled systems. I am the target audience for this manuscript and I have been lost several times while reading it. I think the topic covered in the manuscript is interesting enough to warrant its publication but the following points must be addressed prior to publication:

  • The English language through the paper needs to be reviewed, although it can mostly be understood, in some sections the sentences do not make much sense. For instance, the sentence in lines 38-42. Other similar examples are lines 57-58 or 140-142. Not sure what the authors mean by these statements and they are at best confusing.
  • In page 4, when discussing the construction of the co-assemblies the authors mention the mixing of different ratios and refer to the SI figures S1-S7. Those figures for the different ratios of the two components of the assembly, show several DLS traces. However, do not mention what that traces are. Are they just several measurements of the experiment? If so, this should be specified in the SI.
  • The DLS experiments mentioned in point 2 show that the two moderately stable and fairly monodisperse assemblies are when the components are mixed in 1:2 and 1:2.5 ratios. Since the PDI for the latter ratio is not specified, only says it is low, how feasible is that both mixtures give raise to the same assembly? The authors give different diameters for each assembly, however the associated error to measurement could mean that they both give the same assembly.
  • In the conclusions, lines 369-370, the authors mention that the DNA interaction with the assemblies is completely different than the interaction of the pillar[5]arene with DNA. However previously, in lines 188-189 they mention that the interaction is similar. This seems contradictory and the authors should try to clarify it.

Author Response

  • The English language through the paper needs to be reviewed, although it can mostly be understood, in some sections the sentences do not make much sense. For instance, the sentence in lines 38-42. Other similar examples are lines 57-58 or 140-142. Not sure what the authors mean by these statements and they are at best confusing.

Answer: All the samples kindly provided by Reviewer were substantially re-written. Besides, the manuscript was fully reconsidered to improve the style and remove misprints and technical mistakes.

  • In page 4, when discussing the construction of the co-assemblies the authors mention the mixing of different ratios and refer to the SI figures S1-S7. Those figures for the different ratios of the two components of the assembly, show several DLS traces. However, do not mention what that traces are. Are they just several measurements of the experiment? If so, this should be specified in the SI.

Answer: Usually such traces seen in DLS belong to aggregates formed from the main fraction of particles in the sticking. Their amount is less 1.5%.

  • The DLS experiments mentioned in point 2 show that the two moderately stable and fairly monodisperse assemblies are when the components are mixed in 1:2 and 1:2.5 ratios. Since the PDI for the latter ratio is not specified, only says it is low, how feasible is that both mixtures give raise to the same assembly? The authors give different diameters for each assembly, however the associated error to measurement could mean that they both give the same assembly.

Answer: In the studied variation of the reactant ratio, only two systems, i.e., 1:2 and 1:2.5, produce almost monodisperse systems with PDI 0.18±0.01 in both cases. This indicates that the stoichiometry of interpolyelectrolyte nanoparticles is close to 1:1 (monomolar ratio of charged groups). It can be proposed that these aggregates iate have the same assembly.

  • In the conclusions, lines 369-370, the authors mention that the DNA interaction with the assemblies is completely different than the interaction of the pillar[5]arene with DNA. However previously, in lines 188-189 they mention that the interaction is similar. This seems contradictory and the authors should try to clarify it.

Answer: We delete this sentence (line 369-370): It turned out that the interaction of co-interpolyelectrolytes and with DNA is completely different from the interaction of pillar[5]arene with DNA.

Reviewer 2 Report

This is a well written report that is logically presented and the data soundly analyzed and should be of interest to the readers of Nanomaterials.  With the exception of very few minor grammatical errors, I have only one concern and that is in the analysis of the CD data.  CD spectra are very sensitive to DNA  conformation (B-DNA vs Z-DNA vs DNA quadruplexes vs DNA I-motif) and any changes to the spectra, even minor, could be attributed to a conformational change even if only a slight conformational change. Line 238 suggest that the observe 13 nm shift is consistent with no conformational change in the DNA secondary structure and I would be hesitant to agree with that.

Author Response

Answer: As follows from the literature data, the CT-DNA exerts a positive signal at 275 nm, which corresponds to the base stacking and a negative signal at 245 nm corresponding to right-handed helicity, e.g., B form of the DNA helix [44-46]. According to the Reviewer comment, we have additionally analyzed the literature. The authors of the article (Norden, B.; Kurucsev, T. Analysing DNA complexes by circular and linear dichroism. J. Mol. Recognit. 1994, 7, 141-156. https://doi.org/10.1002/jmr.300070211) believe that it can be a mixture of А и B form of DNA. Therefore, we corrected this part of manuscript:

CD spectrum of CT-DNA has a positive signal at 275 nm, which corresponds to base stacking and a negative signal at 245 nm correspond to right-handed helicity. It is a mixture of A и B form of DNA [44-47]. It is well known that canonical A and B forms of DNA are simultaneously present in aqueous solutions. A-DNA to B-DNA conversion in aqueous solution is quite easy to carry out depending on the environment [48, 49]. In the presence of AP[5]A/STC[4]A and AP[5]A, a 13 nm red shift of the band at 275 nm was observed. These spectral changes confirm the preservation of the canonical form of DNA, however, there is a slight change in the ratio of A and B forms, with an increase in the content of the A form [50]. These changes indicate that CT-DNA retains a B-form upon binding, but at the same time undergoes some conformational changes. Therefore, CT-DNA is bound by ligand (AP[5]A/STC[4]A and AP[5]A) due to electrostatic interactions with phosphate groups. The impossibility of intercalating the studied compounds in DNA confirms the preservation of the shape of the CD curve: the absence of additional bands in the spectrum indicates the non-cleavage of the biopolymer molecule [50], which is in good agreement with the data obtained by fluorescence spectroscopy (Figure 2).

Reviewer 3 Report

This manuscript reports on the preparation of assemblies of ionized macrocycles, namely thiacalix[4]arene and pillar[5]arene, by exploiting their differently charged rims and electrostatic as well as hydrophobic interactions. The idea is not completely new because the Authors have already published on similar assemblies (ref 33). In the present case they have substituted a positively charged calixarene with a pillar[5]arene. The novelty therefore can be justified by this substitution. They also investigated the complexation of the obtained nanoparticles with DNA (also this part resembles previous studies). I found interesting the information inferred from simple techniques such as UV-vis spectrophotometry, CD and fluorescence spectroscopy, DLS and SEM. Nevertheless, I think several points should be clarified/deepened in order to allow its publication.

1) First of all, the English language of the paper needs to be improved. Some sentences are almost nonsensical. For examples “associates”, present all over the manuscript, is not a proper term to indicate the assembly or association.

2) The last sentence of the abstract should be rewritten: it is full of mistakes and typos.

3) Figures S1-S7 are not clear at all. Authors should insert the proper legenda indicating the meaning of the different colour of the curves. In particular, it would be proper to highlight the behaviour of the single components with respect to the association of the two components.

4) The discussion regarding the increased adsorption of the complex-DNA of Figure S8 is not clear at all. Please rewrite the sentence.

5) Despite Figure 5 sketches the combined assembly and self-assembly of pillarene, it would be useful to sketch also the self-assembly of thiacalix[4]arenes, at least to justify the large dimensions of the pure STC[4]A assembly.

6) Moreover, how can the Author justify the cubic assembly of pure pillarenes if both rims are highly positives and therefore repulsive forces may exist between them? I think this part requires a deepening

7) I cannot understand the reasoning of the larger dimensions of the diluted samples (last sentences in the Discussion section). Please clarify this point.

Author Response

1) First of all, the English language of the paper needs to be improved. Some sentences are almost nonsensical. For examples “associates”, present all over the manuscript, is not a proper term to indicate the assembly or association.

Answer: The manuscript was substantially modified, the term assembly was substituted with aggregate

2) The last sentence of the abstract should be rewritten: it is full of mistakes and typos.

Answer: The sentence was changed as follows:

It provides important advantage for the design of effective biomolecular gene delivery systems.

3) Figures S1-S7 are not clear at all. Authors should insert the proper legenda indicating the meaning of the different colour of the curves. In particular, it would be proper to highlight the behaviour of the single components with respect to the association of the two components.

Answer: All the DLS measurements were performed in six repetitions. Zetasizer Nano instrument offers option of multiple measurements to investigate the effect on particle size over time or to prove repeatability. Thus, each line in Figures S1-S7 is an average from six individual  measurements.

4) The discussion regarding the increased adsorption of the complex-DNA of Figure S8 is not clear at all. Please rewrite the sentence.

Answer: We added the explanation in the manuscript:

The baseline increase in the UV spectra was observed during the interaction of the AP[5]A/STC[4]A with the CT-DNA (Figure S8). It is due to the aggregation of the DNA with the AP[5]A/STC[4]A

5) Despite Figure 5 sketches the combined assembly and self-assembly of pillarene, it would be useful to sketch also the self-assembly of thiacalix[4]arenes, at least to justify the large dimensions of the pure STC[4]A assembly.

Answer: The self-assembly of STC[4]A was studied early [Yakimova, L. S.; Gilmanova, L. H.; Evtugyn, V. G.; Osin Y. N.; Stoikov, I. I. Self-assembled fractal hybrid dendrites from water-soluble anionic (Thia)calix[4] arenes and Ag+. J. Nanopart. Res. 2017, 19, 173-183. DOI: 10.1007/s11051-017-3868-9]. In there artice it was shown that the STC[4]A formed spherical micelles at critical micelle concentration (about 1 10-5 M). Therefore, it was decided referring to this information insteqad of ist duplication.

The large dimensions of the pure STC[4]A assembly are possible due to: (1) bulky structure of the macrocycles with internal cavity, (2) presence of four charged groups, (3)  distorted cone conformation caused by electrostatic repulsion of the charged sulfonate groups. These factors lead to the formation of micelles of complex composition. The micelles can form aggregates of different size.

6) Moreover, how can the Author justify the cubic assembly of pure pillarenes if both rims are highly positives and therefore repulsive forces may exist between them? I think this part requires a deepening

Answer: AP[5]A is an organic salt containing ten quaternary ammonium fragments with iodides as counter ions. Its behavior within the crystallization is similar to that of inorganic salts. Thus, AP[5]A can form cube-like aggregates.

7) I cannot understand the reasoning of the larger dimensions of the diluted samples (last sentences in the Discussion section). Please clarify this point.

Answer: We rewrite this sentence:

Probably, when the solution is diluted, the mobility of the polyelectrolyte molecules of the outer layer increases and colloid aggregation takes place (Figure 7).

Reviewer 4 Report

The Authors report the synthesis of polyelectrolyte complexes formed by two macrocyclic structures and investigated their recognition and complexation  ability with calf thymus DNA. The issues reported in this paper have relevance in nanotechnology field and in drug delivery. I believe that the manuscript can be accepted for publication in this Journal after some minor revisions.

The main criticism is represented by the lack of studies regarding the impact of pH changes on the DNA complexation abilities with the positively charged particles. Since the synthesized complexes could be used in gene delivery, the Z-potential analyses should be performed in the pH range of 5.5-8.0 (in order to avoid the premature dissociation of DNA before delivery to cells).

Finally, the English language and style should be improved.

Author Response

The main criticism is represented by the lack of studies regarding the impact of pH changes on the DNA complexation abilities with the positively charged particles. Since the synthesized complexes could be used in gene delivery, the Z-potential analyses should be performed in the pH range of 5.5-8.0 (in order to avoid the premature dissociation of DNA before delivery to cells).

Answer: All the experiments were performed at physiological pH value (pH = 7.4, 10 mM Tris-HCl, 1 mM NaCl). No additional experiments were performed outside the above range (pH  5.5-8.0). However, it was shown that the zeta potentials of the cationic block copolymer micelles were positive with the pH altered by 2 units. This suggests applications utilizing micelle charge for the stabilization can be viable in a wider range of the solution conditions [D. Sprouse, Y. Jiang, J. E. Laaser, T. P. Lodge, T. M. Reineke, Tuning Cationic Block Copolymer Micelle Size by pH and Ionic Strength Biomacromolecules, 2016, 17, 9, 2849-2859].

Finally, the English language and style should be improved.

Answer: We corrected the manuscript.

Round 2

Reviewer 3 Report

The manuscript has improved a lot with respect to the original version. Nevertheless I suggest a few further English amendaments:

1) Page 1 line 13: Abstract: “Self assembly” rather than “self assemble”.

2) Page 2, line 57: Substitute “fixed certain molecules” with “nanomachines”

3) Page 2, line 70: Erase “due to”

4) Page 3 line 98: Change “Measurements were performed at the detection angle of 173° and operated with the software automatically determined the measurement” with “Measurements were performed at the detection angle of 173° with the software automatically determining the measurement”

5) Page 4, line 143: erase “earler” and change it with “in a previous study”

6) Page 5, line 173: Change “corresponded tot he” with “corresponding to the”

7) Page 5, line 180: Erase “a”

8) Page 5, line 209: Erase “that”

9) Page 5, line 210: Change “It can be result of” with “It can be the result of”

10) Page 6, line 212: Change “incorporation” with “incorporate”

11) Page 6, line 226. Change “correspond” with “corresponding”, translate “и“ and change “form” with “forms”

12) Page 7, line 262. Correct “difficulty in restructuring the aggregate” with “difficult restructuring of the aggregate”

13) Page 7, line 320. What does it means “Within them studied”

Finally, in the Supplementary materials. What difference there is between Figure S48 and S49? The concentrations seem the same.

After this modifications I think the manuscript can be published.

Author Response

The manuscript has improved a lot with respect to the original version. Nevertheless I suggest a few further English amendaments:

1) Page 1 line 13: Abstract: “Self assembly” rather than “self assemble”.

2) Page 2, line 57: Substitute “fixed certain molecules” with “nanomachines”

3) Page 2, line 70: Erase “due to”

4) Page 3 line 98: Change “Measurements were performed at the detection angle of 173° and operated with the software automatically determined the measurement” with “Measurements were performed at the detection angle of 173° with the software automatically determining the measurement”

5) Page 4, line 143: erase “earler” and change it with “in a previous study”

6) Page 5, line 173: Change “corresponded tot he” with “corresponding to the”

7) Page 5, line 180: Erase “a”

8) Page 5, line 209: Erase “that”

9) Page 5, line 210: Change “It can be result of” with “It can be the result of”

10) Page 6, line 212: Change “incorporation” with “incorporate”

11) Page 6, line 226. Change “correspond” with “corresponding”, translate “и“ and change “form” with “forms”

12) Page 7, line 262. Correct “difficulty in restructuring the aggregate” with “difficult restructuring of the aggregate”

Answer: We corrected these English amendments.

13) Page 7, line 320. What does it means “Within them studied”

Answer: We delete this phrase.

Finally, in the Supplementary materials. What difference there is between Figure S48 and S49? The concentrations seem the same.

Answer: We corrected concentration. 
